# Distinct Fatty Acid Compositions of HDL Phospholipids Are Characteristic of Metabolic Syndrome and Premature Coronary Heart Disease—Family Study

**DOI:** 10.3390/ijms22094908

**Published:** 2021-05-06

**Authors:** Timo Paavola, Ulrich Bergmann, Sanna Kuusisto, Sakari Kakko, Markku J. Savolainen, Tuire Salonurmi

**Affiliations:** 1Research Center for Internal Medicine, Department of Internal Medicine, Oulu University Hospital and University of Oulu, 90200 Oulu, Finland; timo.paavola@student.oulu.fi (T.P.); sakari.kakko@oulu.fi (S.K.); markku.savolainen@oulu.fi (M.J.S.); 2Medical Research Center Oulu, Oulu University Hospital and University of Oulu, 90200 Oulu, Finland; 3Protein Analysis Core Facility, Biocenter Oulu, University of Oulu, 90570 Oulu, Finland; ulrich.bergmann@oulu.fi; 4Computational Medicine, Faculty of Medicine, Biocenter Oulu, University of Oulu, 90570 Oulu, Finland; sanna.kuusisto@uef.fi; 5NMR Metabolomics Laboratory, School of Pharmacy, University of Eastern Finland, 70210 Kuopio, Finland

**Keywords:** coronary heart disease, metabolic syndrome, HDL subfraction, lipidomics, phospholipid, mass spectrometry

## Abstract

HDL particles can be structurally modified in atherosclerotic disorders associated with low HDL cholesterol level (HDL-C). We studied whether the lipidome of the main phosphatidylcholine (PC), lysophosphatidylcholine (LPC) and sphingomyelin (SM) species of HDL2 and HDL3 subfractions is associated with premature coronary heart disease (CHD) or metabolic syndrome (MetS) in families where common low HDL-C predisposes to premature CHD. The lipidome was analyzed by LC-MS. Lysophosphatidylcholines were depleted of linoleic acid relative to more saturated and shorter-chained acids containing species in MetS compared with non-affected subjects: the ratio of palmitic to linoleic acid was elevated by more than 30%. A minor PC (16:0/16:1) was elevated (28–40%) in MetS. The contents of oleic acid containing PCs were elevated relative to linoleic acid containing PCs in MetS; the ratio of PC (16:0/18:1) to PC (16:0/18:2) was elevated by 11–16%. Certain PC and SM ratios, e.g., PC (18:0/20:3) to PC (16:0/18:2) and a minor SM 36:2 to an abundant SM 34:1, were higher (11–36%) in MetS and CHD. The fatty acid composition of certain LPCs and PCs displayed a characteristic pattern in MetS, enriched with palmitic, palmitoleic or oleic acids relative to linoleic acid. Certain PC and SM ratios related consistently to CHD and MetS.

## 1. Introduction

A low HDL cholesterol level (HDL-C) is a strong established risk factor of coronary heart disease (CHD) [1]. Although Mendelian randomization studies [2,3,4] have shown that the level of HDL-C cannot be considered a causal factor for CHD, plasma HDL consists of various particles with distinct structures and functional properties [5]. The structural characteristics and functionality of these particles may be diversely changed in low HDL-C trait-associated cardiometabolic disorders [6].

The lipidome contributes approximately a half to the total HDL mass, being a key structural component of HDL particles [7]. Ca. 40–60% of the total lipid weight consists of phospholipids and sphingomyelins (SM). These lipids form the surface of spherical HDL particles, acting as an interface between the particles and their surroundings and thus potentially modifying their biological functionality. There are published data about the phospho- and sphingolipidome of HDL in cardiometabolic disorders. The composition of HDL in subjects with extremely low HDL-C displayed lower levels of lysophosphatidylcholines (LPC) [8,9], sphingomyelins [8] and phosphatidylethanolamine plasmalogens [9] than in subjects with extremely high HDL-C. Subjects with CHD had lower levels of certain phosphatidylcholines (PC) as well as of some phosphatidylcholine plasmalogens than healthy subjects [10]. In women with dyslipidemia and type 2 diabetes, HDL was enriched with lysophosphatidylcholines, whereas the contents of the most abundant phosphatidylcholines, phosphatidylethanolamine plasmalogens, sphingomyelins and ceramides were reduced [11]. Metabolic syndrome was associated with high levels of lysophosphatidylcholines and phosphatidylinositols and with low levels of plasmalogens and sphingomyelins in both the HDL2 and HDL3 subfractions [12]. Findings regarding the fatty acid composition of phospholipid species have also been reported. For instance, linoleic acid (18:2) containing species of phosphatidylcholines were reduced in women with type 2 diabetes and dyslipidemia, in type 2 diabetes and in CHD [11,13], whereas PC (16:0/20:3) and PC (18:0/20:3) species were increased in women with type 2 diabetes and dyslipidemia [11].

We performed a lipidomic analysis of HDL subfractions in premature CHD focusing specifically on certain lipid species (PC, SM and LPC) of phospho- and sphingolipids. The study samples were derived from Finnish families with subjects with premature CHD and low HDL-C prior to initiation of statin medication as a major conventional risk factor of the disease. The presence of metabolic syndrome was common in these families. There were also subjects without CHD, metabolic syndrome, low HDL-C or other cardiometabolic disorders in these families. Our hypothesis was that a certain pattern of lipidomic features in HDL subfractions would be characteristic of premature CHD within these families. This pattern would be similarly visible in the lipidome of statin-treated subjects with premature CHD and low HDL-C as its major risk factor as in the lipidome of subjects without clinically manifested CHD or statin treatment but with low HDL-C or metabolic syndrome.

## 2. Results

### 2.1. Study Population

Clinical and biochemical data of the population (112 subjects from 24 families) are shown in Table 1. The population was largely middle-aged and the majority of its CHD patients (*n* = 39) were men. Nearly all CHD patients had premature CHD (age of onset before 60 years of age). Twenty-nine (74%) subjects with CHD also had metabolic syndrome. In addition, some family members without CHD had only metabolic syndrome or low HDL-C (<1.00 mmol/L in men and <1.30 mmol/L in women). CHD was not studied in females since the number of females with CHD was too low (*n* = 8) and the patients were clearly older than women without CHD (median, interquartile range: 67, 60–72 years versus 45, 36–55 years, respectively). Thus, the association between premature CHD and the lipidome of HDL was investigated in men only. Seven men without CHD who were under 35 years of age (there were no premature CHD-affected men under 35 years of age in the study population) were excluded from the statistical comparison to adjust the affected and non-affected groups for age. The excluded subjects were also considered too young to clinically manifest CHD in terms of the strict clinical criteria applied to define CHD in this study, based partly on the observed distribution of the age of CHD onset in the population. Twenty-two (73%) of the 30 premature CHD-affected male subjects included in the statistical comparison also had metabolic syndrome versus ten (50%) of the 20 non-affected male subjects included in the comparison (*p* > 0.05, Table 1). Nearly all (90%) of the premature CHD-affected male subjects were on statin treatment and over half of them (57%) used the angiotensin-converting enzyme inhibitor or angiotensin receptor II-blocking medication, the usage whereof was minor (5% in both cases) in the non-affected group. The premature CHD-affected male subjects also displayed heavier exposure to smoking than the non-affected subjects. When comparing subjects with or without metabolic syndrome, the affected subjects were older than the non-affected subjects. The affected subjects also used the angiotensin-converting enzyme inhibitor or angiotensin receptor II-blocking medication more frequently than the non-affected subjects. Men affected by metabolic syndrome were more exposed to smoking than the non-affected men (median, interquartile range: 22, 10–32 pack-years versus 5, 0–13 pack-years, respectively, *p* < 0.05). At blood sample collection, 42 subjects altogether in the study population were using statins and 36 were using angiotensin-converting enzyme inhibitors or angiotensin receptor II blockers. The lipidome of HDL was also investigated in a subpopulation where statin-treated or CHD-affected subjects were excluded. This subpopulation showed a similar clinical and biochemical profile as the whole population (Table 1).

### 2.2. Lipidomic Associations of HDL Fractions with Metabolic Syndrome, HDL-C and Premature CHD

Statin treatment is known to elevate HDL-C mildly [14], but there are not much data about the effect of statin therapy on the molecular phospholipid composition of HDL. In the Watanabe rabbit hyperlipidemic animal model, pitavastatin (lipophilic statin) therapy has been shown to decrease the total serum lysophosphatidylcholine level (e.g., the LPC 16:0 level) and to increase the total sphingomyelin content in the LDL fraction and, to some extent, in serum [15]. Statin medication reduces the production of lipoprotein-associated phospholipase A2 which hydrolyzes oxidized phospholipids into lysophosphatidylcholines [15,16]. In Watanabe rabbits, pitavastatin therapy also increased the levels of n-6 fatty acid-containing phosphatidylcholines, e.g., of PC (18:0/20:3) and PC (18:0/20:4), and pitavastatin therapy similarly elevated PC 38:4 and PC (16:0/20:4) levels variably in HDL2 and HDL3 fractions in a human study [17]. Similar characteristics are observed in our data (Figure 1, Appendix A). For these potential confounding effects of statin medication, we do not emphasize findings which were detected only in respect to statin-treated CHD. The findings which were connected only with metabolic syndrome are reported, although they did not show any relation to statin-treated premature CHD. All the results of statistical testing can be seen in Appendix A. Specifically, we report only the lipidomic differences that were significant in respect (A) to metabolic syndrome although not showing any significant relation to statin-treated premature CHD or in respect (B) to metabolic syndrome and/or related to HDL-C and consistently (similarly decreased/increased) significant in respect to statin-treated premature CHD. By the phrase ‘consistently’ we hypothesize that metabolic syndrome, premature CHD and low HDL-C are biologically interrelated phenotypes in this population. Thus, for example, a lipid variable which was higher in premature CHD-affected than in non-affected subjects would presumably be higher in metabolic syndrome-affected than in non-affected subjects, as well as correlate inversely with HDL-C. The associations of lipid parameters with metabolic syndrome (points A and B) or HDL-C alone (point B) were confirmed by excluding statin-treated or CHD-affected subjects from the analysis. Results of these confirmatory analyses are shown in Appendix A. There were only modest sex differences in some of the lipid parameters (average ± standard deviation of the absolute value of percentage difference between men and women = 7.3 ± 6.4%). Age was not commonly associated with lipid parameters (average ± standard deviation of proportion of individual lipid parameter variance explained (R2) = 4.2 ± 4.8%). All the analyses were adjusted for sex and age.

### 2.3. Fatty Acid Composition of Lysophosphatidylcholine and Phosphatidylcholine Species in Metabolic Syndrome

Lysophosphatidylcholines of HDL showed a distinctive profile in metabolic syndrome. The affected subjects exhibited lower levels of diunsaturated linoleic acid-containing lysophosphatidylcholines than subjects without metabolic syndrome, whereas the levels of saturated or monounsaturated fatty acid-containing lysophosphatidylcholines were statistically non-significantly higher than or similar to in non-affected subjects (Figure 1). The LPC 18:2 level was approximately 20% lower in affected subjects, although the difference dropped below the limit of significance in the confirmatory analysis where statin-treated and CHD-affected subjects were excluded (Appendix A). Diunsaturated LPC 18:2 was also longer chained than saturated LPC 16:0 whose unsaturated forms were not detected in this study.

Consistently with the findings regarding fatty acid saturation and chain length of individual lysophosphatidylcholines, their ratios exhibited a more characteristic and significant pattern in metabolic syndrome. Subjects with metabolic syndrome had a significantly lower amount of diunsaturated linoleic acid-containing lysophospholipids relative to saturated stearic acid-containing lysophosphatidylcholines than subjects without metabolic syndrome in their HDL: a ratio of LPC 18:0 to 18:2 in both HDL2 and HDL3 fractions was higher in affected subjects (Figure 1). Subjects with metabolic syndrome also had higher amounts of shorter-chained saturated palmitic acid containing lysophosphatidylcholines compared to longer-chained saturated stearic acid containing lysophosphatidylcholines in their HDL: the ratio of LPC 16:0 to 18:0 was higher in affected subjects in HDL2 and HDL3 fractions alike (Figure 1). Illustrating a combined effect of the degree of saturation and the chain length of lysophosphatidylcholine fatty acids, the ratio of LPC 16:0 to 18:2 was over 30% higher in subjects with metabolic syndrome than in non-affected subjects in both HDL fractions (Figure 1, Appendix A). To elucidate these findings, associations between the abovementioned ratios and various metabolic and clinical parameters connected with metabolic syndrome (HDL-C, plasma total triglycerides, VLDL triglycerides, VLDL protein, total adiponectin, HMW adiponectin, waist circumference and HOMA-IR) were analyzed. A high LPC 16:0 to 18:2 ratio was associated with an adverse metabolic and anthropometric profile with elevated total plasma triglycerides, VLDL triglycerides and VLDL protein levels, as well as with high waist circumference in HDL2 and HDL3 fractions alike (Appendix A).

Together with sphingomyelins, phosphatidylcholines formed the most abundant lipid class measured in this study. In both fractions, the most substantial of them were 34 or 36 fatty acid carbon atoms containing molecules which carried palmitic (16:0) or stearic (18:0) and linoleic (18:2) fatty acids verified by MS/MS. In metabolic syndrome, linoleic acid-containing PC (16:0/18:2) and PC (18:0/18:2) levels in the HDL3 fraction were 12% and 15% lower, respectively (Appendix A, Figure 1). Only PC (16:0/18:2) showed significant associations with parameters of metabolic syndrome in the confirmatory analysis where statin-treated and CHD-affected subjects were excluded (Appendix A). Ratios of oleic acid- (18:1) to linoleic acid-carrying species of the 34 and 36 fatty acid carbon atom-containing phosphatidylcholines were also associated with metabolic syndrome: the ratio of PC (16:0/18:1) to PC (16:0/18:2) was elevated in both HDL fractions in subjects with metabolic syndrome (11% in HDL2 and 16% in HDL3, Appendix A, Figure 1) and the ratio of PC (18:0/18:1) to PC (18:0/18:2) was increased by 18% in their HDL3 fraction only (Appendix A, Figure 1). The ratio of PC (16:0/18:1) to PC (16:0/18:2) was positively related to total plasma triglycerides, VLDL triglycerides and VLDL protein level in both fractions, as well as to waist circumference in the HDL3 fraction, similarly to the ratio of PC (18:0/18:1) to PC (18:0/18:2) in the HDL3 fraction (Appendix A).

A minor palmitoleic acid-containing PC (16:0/16:1) was explicitly related to metabolic syndrome (Figure 1) where affected subjects displayed a 40% higher level in the HDL2 fraction and a 28% higher level in the HDL3 fraction (Appendix A). The phospholipid level in both fractions was also associated with plasma levels of total and VLDL triglycerides, as well as with VLDL protein and waist circumference (Appendix A). Another minor PC 35:2 was significantly decreased in metabolic syndrome only in the HDL3 fraction but exhibited consistent associations with the parameters of metabolic syndrome in both fractions (Figure 1, Appendix A).

### 2.4. Phosphatidylcholines PC (18:0/20:3) and PC (18:0/20:4) in Metabolic Syndrome and Premature CHD

Two phosphatidylcholine species with a relatively high total carbon number, multiple double bonds in their fatty acids and modest levels in samples exhibited findings worth being highlighted. First, the PC (18:0/20:3) level was positively associated with premature CHD (Figure 1), being 27% higher in the HDL2 fraction and 12% higher in the HDL3 fraction in subjects with premature CHD, but showed no difference in respect to metabolic syndrome in the confirmatory analysis where statin-treated and CHD-affected subjects were excluded (Appendix A). When proportioned to the levels of abundant linoleic acid-carrying PC (16:0/18:2) and PC (18:0/18:2) which were linked inversely with metabolic syndrome in the HDL3 fraction (see section above), the difference of these ratios remained highly significant in both HDL fractions and in respect to both premature CHD and metabolic syndrome: the decreases of the ratios of PC (16:0/18:2) to PC (18:0/20:3) and of PC (18:0/18:2) to PC (18:0/20:3) ranged from 14% to 25% in the HDL2 fraction and from 15% to 22% in the HDL3 fraction in affected subjects (Appendix A, Figure 1). The ratios were also consistently associated with many of the parameters of metabolic syndrome (Appendix A).

The level of another phospholipid with similar chain length, degree of saturation and relative level in the samples, PC (18:0/20:4), was also 27% higher in the HDL2 fraction and 15% higher in the HDL3 fraction in premature CHD (Appendix A, Figure 1) but showed no difference in respect to metabolic syndrome or was not related to HDL-C. Its ratios to the linoleic acid-containing species, PC (16:0/18:2) to PC (18:0/20:4) and PC (18:0/18:2) to PC (18:0/20:4), were 17–24% lower in premature CHD in both HDL fractions (Appendix A, Figure 1). The PC (16:0/18:2) to PC (18:0/20:4) ratio in the HDL3 fraction was consistently related to HDL-C (stand. beta = 0.25, Figure 1), including in the confirmatory analysis where statin-treated or CHD-affected subjects were excluded, but showed few associations with the other parameters of metabolic syndrome (Appendix A).

### 2.5. Sphingomyelins in Metabolic Syndrome and Premature CHD

Subjects affected by metabolic syndrome exhibited lower levels of many sphingomyelins than non-affected subjects (Appendix A), whereas men with premature CHD (mostly statin-treated) generally had higher levels of sphingomyelins than men without CHD (mostly non-statin-treated). The ratios of sphingomyelins were therefore investigated to find common features in respect to both metabolic syndrome and statin-treated premature CHD. Only total carbon atom and double bond numbers of these molecules are reported. SM 34:1, an abundant short-chained species, can be SM (d18:1/16:0) based on its abundance (7).

Ratios of certain less abundant sphingomyelin species to particular more abundant species were increased in subjects affected by premature CHD or metabolic syndrome. The ratio of SM 36:2 to SM 34:1 displayed the strongest association with cardiometabolic disorders in both HDL fractions (Figure 1). It was increased by 36% in the HDL2 fraction and by 17% in the HDL3 fraction in subjects affected by premature CHD and related consistently to HDL-C in the HDL2 fraction (stand. beta = −0.34) and the HDL3 fraction (stand. beta = −0.41, Appendix A). The ratio of SM 36:2 to SM 34:1 in the HDL2 fraction was also associated with many other parameters of metabolic syndrome (Appendix A). There were also some ratios which were related to premature CHD or metabolic syndrome in the HDL2 fraction only. The ratio of SM 36:2 to SM 42:3 in the HDL2 fraction was 31% lower in premature CHD and 27% lower in metabolic syndrome and related to HDL-C (stand. beta = −0.39, Appendix A, Figure 1) as well as to other parameters of metabolic syndrome (Appendix A).

### 2.6. Lipidomic Associations of Premature CHD Adjusted for Metabolic Syndrome Status, HDL-C, Other Parameters of Metabolic Syndrome or Cardiovascular Risk Factors

Here, we report whether the above-presented lipidomic parameters that were consistently associated both with premature CHD and metabolic syndrome or with premature CHD and low HDL-C were significantly related to premature CHD after adjusting the analyses separately for metabolic syndrome status or HDL-C (Table 2). The parameters tested were the following PC and SM ratios: PC (16:0/18:2) to (18:0/20:3), PC (18:0/18:2) to PC (18:0/20:3), PC (16:0/18:2) to PC (18:0/20:4) and SM 36:2 to SM 34:1 and SM 36:2 to SM 42:3. We found that many of these ratios displayed significant associations with statin-treated CHD in the adjusted analysis as well. In the HDL2 fraction, all the PC ratios and the SM 36:2 to SM 34:1 ratio remained associated with premature CHD in all the adjusted analyses (unadjusted *p* < 0.01). In the HDL3 fraction, the SM ratios were not significantly related to premature CHD in the adjusted analyses and the PC ratios fell below the limit of significance when adjusted for HDL-C or for some other parameters of metabolic syndrome. Thus, the ratios of HDL2 fraction phosphatidylcholines and sphingomyelins displayed more significant adjusted associations with premature CHD than the ratios of the corresponding lipids in the HDL3 fraction.

### 2.7. Phospholipids and Serum Free Fatty Acids

Finally, we conducted a correlation analysis to find out how the measured lipids and their ratios are associated with the serum free fatty acid level (Appendix A). Elevated serum free fatty acids are a marker of lipolysis and can be detected, e.g., in diabetes [18]. Statin medication has also been thought to exert part of its protective effects by impacting plasma free fatty acid concentrations [18]. Indeed, the positive correlations between serum free fatty acids and lysophosphatidylcholines and the negative correlations between serum free fatty acids and PC (18:0/20:3) or PC (18:0/20:4) are opposite to the associations between statin medication and those lipid levels (see Section 2.2). Moreover, lysophosphatidylcholines are products of fatty acid removal, similarly to free fatty acids in serum, which could be a mechanism behind their direct relation.

## 3. Discussion

This study showed that the lipidome of PCs, LPCs and SMs in HDL subfractions contained many individual lipids associated with cardiometabolic disorders. As prominent examples of individual lipidomic features identified in this study, certain compositions of lysophosphatidylcholine fatty acids were characteristic of metabolic syndrome; the ratio of LPC with palmitic acid (16:0) to LPC containing diunsaturated linoleic acid (18:2) was over 30% higher in affected than in non-affected subjects. Of note, explicitly (approximately 30–40%) higher levels of monounsaturated palmitoleic acid-containing PC (16:1/16:0) were detected in metabolic syndrome as well. Some fatty acid compositions of phosphatidylcholines were consistently linked with premature CHD and metabolic syndrome; for example, the ratio of an abundant linoleic acid-containing PC (16:0/18:2) to PC (18:0/20:3) was approximately 15–25% lower in metabolic syndrome or premature CHD than in non-affected subjects. The differences in fatty acid compositions were mostly detected in both HDL fractions. Many of our findings have not been reported or emphasized previously in mass spectrometric HDL lipidome studies published earlier. However, some of our findings confirm previously published lipidomic data on cardiometabolic disorders as discussed next. Before doing so, we would like to emphasize that the focus of this study was the clinically relevant CHD patient group with low HDL-C. The premature CHD-affected subjects had low HDL-C as a major risk factor of the disease and had many other conventional cardiovascular risk factors (e.g., a high LDL cholesterol level) within specified limits prior to statin treatment.

There were certain limitations in this study. Almost all subjects affected by premature CHD were on statin treatment, which exerts an impact on the lipid composition of HDL (see Section 2.2). Statin-treated premature CHD-affected subjects had higher levels of many sphingomyelins than subjects without CHD. By contrast, metabolic syndrome was associated with generally lower sphingomyelin levels in our study. This inconsistency can be attributed to a modifying effect of statin medication on the sphingolipid content of HDL as CHD patients without any hypolipidemic medication exhibited lower total HDL sphingomyelin content than healthy subjects [19]. The statin-treated premature CHD-affected subjects in our study did not exhibit a similar pattern in lysophosphatidylcholine levels as was observed in respect to metabolic syndrome. In the HDL3 fraction, all their lysophosphatidylcholine levels were significantly lower than in non-affected subjects. Again, a confounding effect of statin medication on lysophosphatidylcholines of the HDL fractions (especially on that of HDL3) is plausible. On the other hand, it is the current standard clinical practice to administer statins to CHD patients, and this kind of study addresses the relevant question of how the lipidome of these patients is modified regardless of statin medication. Another limitation was that the subjects affected by premature CHD included in the analysis were all men. Furthermore, the HDL fraction samples were stored frozen at −70 °C prior to analysis and were thus not freshly isolated. However, our own experimental data showed that the levels of the reported molecular lipid species were not significantly affected by the freezing–storage (months)–thawing–extraction–storage (months) cycle compared with freshly isolated and measured HDL fraction samples (unpublished data). Our sample size was too small to study the association between premature CHD and lipid parameters by adjusting for all the measured established cardiovascular risk factors combined in a single model in a statistically robust way. Compared with a conventional case–control setting, this kind of family study setting may result in underestimation of some lipidomic differences between cases and controls since the control and case groups shared some common genetic family background, plausibly relating to increased CVD risk, as well as to cardiometabolic disorders such as metabolic syndrome or low HDL-C. Finally, lipoprotein oxidation is a key pathophysiological factor in the development of atherosclerosis [20]. Smoking is a well-known risk factor of oxidative stress [21] and the CHD-affected subjects in this study were more exposed to smoking than non-affected subjects (Table 1). Diet can also modify the fatty acid composition of certain HDL lipid classes, such as phosphatidylcholines, in the short term, whereas other classes, such as sphingomyelins, may remain more stable during dietary changes [22]. Dietary supplements may modify the composition of plasma lipoprotein fractions and lipoprotein metabolism [23], but their usage was screened at the recruitment phase of this cross-sectional study (see Section 4.1 ‘Study Subjects’).

Our results show consistency with earlier HDL lipidomic studies performed on subjects affected by similar cardiometabolic disorders as the subjects in our study population. In line with our findings, linoleic acid-containing PC (16:0/18:2) and PC (18:0/18:2) were decreased in women with type 2 diabetes and dyslipidemia [11], in type 2 diabetes and in CHD patients [13]. The proportion of PC 36:2 of total phospholipids was decreased in both HDL2 and HDL3 fractions in metabolic syndrome [12]. In an HDL lipidomic study where CHD patients were investigated, their PC 34:2 and PC 36:2 levels among some odd-chained PCs, such as PC 35:2, were lower in stable CHD compared with controls, whereas their PC 38:4 level was elevated and PC 38:3 showed no difference. Levels of 20:3-containing PC (18:0/20:3) and PC (16:0/20:3) were increased in women with type 2 diabetes and dyslipidemia in the study of Ståhlman et al. [11]. In type 2 diabetes and in metabolic syndrome, the levels of LPC 18:1 and 18:2 were decreased and the level of LPC 16:0 was increased [11,13], but this difference was not detected in CHD patients, over 92% of whom were treated with statin, as in our study [13]. Subjects affected by metabolic syndrome also displayed a significant increase of the LPC 16:0 proportion in their HDL3 fraction of total lysophosphatidylcholines, whereas the LPC 18:1 proportion was decreased compared with controls and the decrease of the LPC 18:2 proportion was non-significant [12]. In the HDL2 fraction, no similar findings were reported. In our study, most of the sphingomyelin species were reduced in metabolic syndrome. This result is in line with previous findings: women with type 2 diabetes and dyslipidemia [11] and subjects with low HDL-C [8], metabolic syndrome [12] or CHD [19] presented decreased total sphingomyelin content in their HDL fractions. The fatty acid composition of sphingomyelins was not confirmed in our study by MS/MS, but the ratios of individual sphingomyelin species in our study (see Figure 1) show some consistency with what has been found in women with type 2 diabetes and dyslipidemia [11] or in subjects with metabolic syndrome [12]. In our study, the SM 42:3 level of the HDL2 fraction in metabolic syndrome was similarly decreased (Appendix A) as in HDL of type 2 diabetes patients [13] and in healthy subjects after a fast food diet [22]. PC (16:0/16:1) and PC (16:0/18:1) levels were elevated in the HDL isolated from subjects under postoperative acute-phase response [24]. Metabolic syndrome is associated with dysregulated inflammatory mechanisms [25] and was also associated with explicitly elevated levels of PC (16:0/16:1) in both HDL fractions in our study [10].

The observed lipidomic features of HDL fractions also bear resemblance to the total plasma lipidome in cardiometabolic disorders described in other studies. When a large number of subjects (*n* = 1358) from Mexican-American families were investigated, it was observed that PC 32:1 and PC 38:3 were higher in subjects with metabolic syndrome [26]. Prediabetes or type 2 diabetes or both presented some similar findings as in our study compared with controls of normal glucose tolerance, such as the elevation of the PC 32:1 and decrease of the 35:2 plasma levels [27]. In that study, plasma levels of lysophosphatidylcholine species exhibited a similar profile as in our study in that the LPC 16:0 and LPC 18:0 levels were increased and the LPC 18:2 level was decreased in affected subjects. Weight loss intervention resulted in the decrease of the total triglycerides level in plasma, predominantly visible in short-chained saturated fatty acids containing triacylglycerols [28]. Subjects affected by stable CHD displaying many similar metabolic and clinical features as the subjects in our study had higher PC 32:1, PC 38:3 and PC 38:4 levels and lower PC 34:2, PC 36:2, PC 33:2 and PC 35:2 levels, whereas plasma levels of the corresponding lysophosphatidylcholine species as measured in our study were generally lower in affected subjects [29]. Plasma PC 38:3, 38:4 and 40:6 levels were the only lipids among the phosphatidylcholines, sphingomyelins, phosphatidylethanolamines and lysophosphatidylcholines measured whose levels correlated with HDL-C not positively but negatively [30]. However, there are some contradictory results, e.g., that the PC 38:3 level was decreased in obesity [31]. In that study, LPC 16:0 and LPC 18:0 levels were positively related to insulin resistance and to an adverse metabolic phenotype associated with visceral obesity. On the contrary, low plasma LPC 16:0 and LPC 20:4 levels and the high SM 38:2 level were associated with an increased risk of prospective adverse cardiovascular outcomes after adjusting for Framingham risk factors (although with high false detection rate values) [32]. In general, a high degree of saturation in plasma lipid fatty acids is associated with an adverse cardiovascular phenotype [33].

In part, the similarities in the results of our HDL phospholipids and plasma phospholipids in other studies exist simply because HDL particles are substantial carriers of total plasma phospholipids and total sphingolipids [23,30]. On the other hand, the similarities reflect metabolism of HDL where its lipids partly originate from other lipoprotein fractions and partly efflux from various cells and tissues either as nascent HDL or incorporating into more mature particles [23,34]. Some of the lipidomic features detected in HDL fractions are also visible in the lipidome of adipocyte membranes of obese twins compared with their lean genetically identical co-twins, such as decreased PC 34:2 and PC 36:2 levels, increased levels of 16:1, 20:3 and 20:4 fatty acids and the decreased level of linoleic acid [31]. The mechanism behind these observations may lie in the altered regulation of fatty acid metabolism in acquired obesity [31]. A similar pattern of total plasma fatty acids predicted metabolic syndrome development during a 20-year follow-up in a 50-year-old male population cohort (*n* = 1558) and was associated with changes in fatty acid metabolism, some of which were independent of lifestyle factors (smoking, BMI and physical activity), whereas others were not [35]. Increased risk of metabolic syndrome was associated with a decreased proportion of linoleic acid and increased proportions of palmitic, palmitoleic and oleic acids as well as 18:3 and 20:3 fatty acids of total serum fatty acids at the baseline. Serum concentrations of linoleic, palmitic, palmitoleic and 20:3 fatty acids were consistently associated with markers of subclinical inflammation in a young healthy adult population [36]. Plasma lipidome may also be an intermediate phenotype between genetic factors and their clinical metabolic outcomes, making some lipidomic features and the links between them potentially genetically determined [37]. Of note, in that study, LPC 16:0 was grouped into a genetically correlated cluster of lipids which was associated with an adverse metabolic phenotype of raised serum triglycerides, whereas LPC 18:2 belonged to a genetic cluster inversely associated with central obesity and metabolic syndrome.

The similarity with the plasma lipidome raises the question of whether the HDL lipidome has independent pathogenetic relevance or whether it is only a reflection of potentially more relevant global lipidomic patterns and metabolic dysregulation. This question cannot be answered based on our data. There are previously published data implying that the observed changes may have an effect on the functionality of HDL particles. For example, phospholipid surface fluidity of recombinant HDL has been shown to regulate its cholesterol efflux capacity and it was modulated by fatty acid chain saturation of phosphatidylcholines [38]. Sphingomyelin has also been shown to modulate the surface properties of HDL [39,40] and its cholesterol efflux capacity [41]. Specifically, both phosphatidylcholine fatty acid saturation and sphingomyelin content modulated cholesterol efflux to and uptake from recombinant HDL in a complex manner, interacting with each other [42]. Moreover, the phosphatidylcholine composition of reconstituted HDL affected its capacity to inhibit expression of adhesion molecules in endothelial cells, where PC (16:0/18:2) containing particles were more potent than PC (16:0/18:1) carrying ones [43]. SM 42:3 has been shown to inhibit apoptosis in vitro [13]. A potential connection between the lipidomic profile, e.g., the relative depletion of linoleic acid-containing phospholipids, of HDL in subjects with metabolic syndrome in our study and the compromised antioxidative capacity of HDL in obesity [44] remains an intriguing question. Increased oxidative stress and cardiovascular burden have been linked with obesity and male gender [45,46].

## 4. Study Subjects, Materials and Methods

### 4.1. Study Subjects

The study population (*n* = 112, Table 1) consisted of Northern Finnish families (*n* = 24), all of which included a proband with premature CHD and low HDL-C as described earlier [47]. The inclusion criteria for probands were HDL-C below 1.0 mmol/L, plasma LDL cholesterol level below 4.0 mmol/L, plasma triglyceride level below 3.0 mmol/L, no treatment for diabetes and the first CHD event (acute myocardial infarction, coronary angioplasty or coronary bypass operation) before the age of 60. All the inclusion lipid criteria had to be fulfilled in the probands before the onset of possible statin treatment. In addition to probands, their biological family members were recruited. Subjects with a clinical manifestation of CHD not fulfilling the previously mentioned strict criteria for CHD were defined as unknown with respect to their CHD status (two subjects). Metabolic syndrome was defined based on the IDF criteria [48]. The subjects were sampled between 2007 and 2009 at the Oulu University Hospital. Medications, nutritional supplements and other relevant medical information was collected using questionnaires and interviews and by reviewing patient records. A written informed consent was obtained from the subjects. The study was approved by the Ethics Committee of the Northern Ostrobothnia Hospital Region and it abides by the Declaration of Helsinki principles.

### 4.2. Materials and Methods

#### 4.2.1. Clinical Measurements

Information regarding smoking history, alcohol intake and use of medications was obtained using a questionnaire. The lifetime smoking burden was calculated as pack-years (pack-year = 20 cigarettes smoked every day during one year) and alcohol intake was expressed as doses/week (dose = 12 g). Waist circumference, height, weight and blood pressure were measured and body mass index was calculated.

#### 4.2.2. Blood Samples

Blood samples were obtained by appointment after an overnight fast (minimum of 8 h) from the forearm in a sitting position. In patients with acute myocardial infarction or coronary bypass operation, blood samples were taken at least three months after the incident. EDTA plasma and serum samples were obtained after centrifugation at 1500× g for 15 min at +4 °C.

#### 4.2.3. Clinical Chemistry Measurements

Serum insulin and plasma glucose, total cholesterol, LDL cholesterol, HDL-C, triglyceride, free fatty acid and creatinine concentrations and alanine transaminase activity were analyzed with Advia (2400/Centaur) automated chemistry analyzers (Bayer Healthcare AG, Leverkusen, Germany) in NordLab Oulu, laboratory of the Oulu University Hospital. Insulin resistance was calculated as the homeostatic model assessment index (HOMA-IR), i.e., (serum insulin [mU/L] × plasma glucose [mmol/L])/22.5 [49].

#### 4.2.4. Adiponectin Measurements

Total adiponectin and high-molecular-weight (HMW) adiponectin were measured using a human adiponectin ELISA kit (Cat # EZHAPD-61K) and a human HMW adiponectin ELISA kit (Cat # EZHMWA-64K) supplied by Linco Research Inc., St. Charles, MO, USA).

#### 4.2.5. Isolation and Analysis of Chemical Composition of Lipoprotein Fractions

Plasma VLDL (d < 1.006 g/mL), LDL (1.019 < d < 1.063 g/mL), total HDL (1.063 < d < 1.210 g/mL), HDL2 (1.063 < d < 1.125 g/mL) and HDL3 (1.125 < d < 1.210 g/mL) were isolated by sequential ultracentrifugation [50] using a Beckman ultracentrifuge (Ti 50.4 rotor, 259,000× g at +15 °C to isolate total HDL; and Ti 50.2 rotor, 218,000× g at +15 °C to isolate HDL2 and HDL3). The isolated HDL fractions were immediately dialyzed against the PBS and stored at −70 °C prior to extraction and lipidomic analysis. Isolated lipoprotein fractions were analyzed for lipids as described [51]. To obtain actual plasma concentrations of lipoprotein-associated lipids, the loss of lipoproteins in the ultracentrifugation procedure was corrected as described [52].

#### 4.2.6. Lipidomic Analysis

HDL fraction samples were extracted according to a protocol modified from Yetukuri et al. [8]. The samples had not been previously thawed. Briefly, 20 µL of a vortexed HDL2 or HDL3 fraction sample, 100 µL of the chloroform/methanol (2:1, *v/v*) solvent and 10 µL of the internal standard mixture containing 11 different lipid compounds (approximately 90 µg/mL of each compound) were mixed. The mixture was shaken for 1 min, extracted for 30 min in the dark and centrifuged (9400× *g*, 3 min). Sixty µL of the lower phase were put into a glass vial with a sealed cap for storage at −20 °C in the dark until analyzed. As the samples were analyzed in multiple batches on the instrument, at least one blank and one HDL control sample were included in every batch for quality control. The HDL control samples were extracted from a pooled sample of the total HDL fraction isolated from healthy donors. They were extracted similarly to the patient samples with corresponding volume ratios but in larger quantities to be divided between multiple glass vials to be analyzed in separate batches. When extracting blank controls, no HDL samples but only their PBS–EDTA buffer solution and the internal standard mixture were extracted. All the patient samples were extracted as duplicates. The internal standard mixture contained PC(17:0/17:0), PC(17:0/0:0) (= LPC 17:0), PE(17:0/17:0), PG(17:0/17:0), Cer(d18:1/17:0), PS(17:0/17:0), PA(17:0/17:0) and sphingosine-1-phosphate (C17 base) from Avanti Polar Lipids, Alabaster, AL. USA), as well as monoheptadecanoin MG(17:0/0:0/0:0), diheptadecanoin DG(17:0/17:0/0:0) and triheptadecanoin TG(17:0/17:0/17:0) from Larodan Fine Chemicals, Malmö, Sweden).

The extracted lipids were analyzed on a Q-ToF (quadrupole time of flight) instrument (Waters Synapt G1) in the MS sensitivity (V) mode recording 0.2 s scans from m/z 100–1500 in the centroid mode. Measurements in the positive and negative modes were taken with tune settings optimized to obtain maximum sensitivity for each polarity. In both polarities, spectra were lock mass corrected using leu-encephalin. Both polarity measurements were performed with the following chromatography method using a Waters Aquity ultraperformance liquid chromatography (UPLC) system with an Aquity T3 2.1 × 100 mm column operated at 55 °C. Eluent A was made with 40% acetonitrile and 60% water, after which ammonium acetate and NH3 solution (density, 0.88 g/mL) were adjusted to 10 mmol/L and 0.01% v/v, respectively. Eluent B comprised 10% acetonitrile and 90% propanol-2, containing the same NH3 and ammonium acetate concentrations as A. The gradient was operated at 0.35 mL/min, changing in the linear mode from 60% B to 100% B in 10 min, where it was held for 2 min and returned to starting conditions in 3 min, allowing for another 2 min equilibration before the next injection.

Lipid samples were dried in a speed vac without heating and dissolved in a 100-µL solvent made from 25 µL chloroform/methanol (1:2, v/v) and 75 µL ethanol for samples intended for positive polarity and in a 25-µL solvent composed of 5 µL chloroform/methanol (1:2, v/v) and 20 µL ethanol for negative polarity. In addition, the solvents contained a second, so called external standard, which was TG (16:0/16:0/16:0-13C3) (50 µg/mL, Larodan Fine Chemicals, Malmö, Sweden) for positive polarity, and tetramyristyl cardiolipin (80 µg/mL) for negative measurements. The purpose of the external standards was to assess errors coming from instrument and method fluctuations. For positive and negative polarity, 4 µL and 5 µL, respectively, were injected.

The data were processed with the Marker Lynx option of the Mass Lynx software suite (Waters) with parameters adjusted to identify compounds (markers) well above the noise level, using mass extraction windows of 0.1 Da and chromatographic peak width of 0.25 min. For quantification, each lipid class was normalized to its class-specific internal standard assuming linear response (PCs relative to PC (17:0/17:0), LPCs relative to PC (17:0/0:0) and SMs relative to Cer (d18:1/17:0)). Lipids were identified first by accurate mass (accuracy better than 5 ppm) and chromatographic elution (comparison to standards). For unclear cases, additional measurements in the MS/MS mode were carried out to verify lipid class and, if possible, fatty acid composition. For downstream analysis, positive polarity measurements were used for PCs, LPCs and SMs. The data obtained by Marker Lynx were manually inspected and, if necessary, overruled by manual processing of extracted ion chromatograms.

The separately extracted patient duplicates were analyzed in different batches. The concentrations of these duplicate samples were averaged to obtain lipid concentration, which was expressed relative to the total protein mass concentration of each sample to obtain the final lipid content. The final content should be interpreted as relative but not absolute concentration. The reported lipid species were not detected in blank control samples. The Average CV% between batches of the reported lipids in HDL control samples was 11%. The following lipids had CV% over 20% in HDL control samples: HDL2: SM 42:1, 27%; SM 34:2, 24%; and HDL3: PC (16:0/18:2), 29%; SM 40:1, 23%; SM 42:1, 26%; SM 42:2, 22%; SM 41:2, 24%; SM 41:1, 29%; PC 35:2, 21%.

#### 4.2.7. Statistical Analysis

Statistical analysis was conducted with the SPSS software (IBM SPSS Statistics for Windows, Version 21, Armonk, NY, USA). The Shapiro–Wilk test and visual inspection were used to assess normality and skewed variables were log- or square root-transformed to obtain the normal distribution. The generalized estimating equation model [53,54] was used to adjust statistical analyses for the fact that study subjects deriving from the same biological family are biologically related to each other and thus observations are non-independent. It was used to analyze differences of lipid parameters in respect to clinical groups as well as associations between lipid variables and continuous variables. The Student’s *t*-test, Mann–Whitney U-test, Pearson’s chi-squared test and Fisher’s test were used to show differences in baseline characteristics. To adjust for multiple testing, the Benjamini–Hochberg procedure was applied, with *p*-values below the false detection rate cutoff of 0.05 considered statistically significant. The total number of hypotheses in a test family was 142, corresponding to the total number of measured lipid levels and their calculated ratios (Appendix A); for example, associations between PC (16:0/18:2) and metabolic syndrome and between the same PC (16:0/18:2) and HDL-C were not included in the same test family as metabolic syndrome and HDL-C are interrelated variables and thus the comparisons cannot be considered truly independent. Multiple testing adjustment was not applied when analyzing adjusted associations between lipid parameters and premature CHD status (Table 2) where the unadjusted false detection rate *p* < 0.01 was considered statistically significant.

## 5. Conclusions

A characteristic fatty acid pattern of lysophosphatidylcholines and phosphatidylcholines in the lipidome of HDL fractions was detected in metabolic syndrome. In addition, certain PC and SM ratios were related to metabolic syndrome and/or low HDL-C and statin-treated CHD in a consistent manner. The CHD patients in this family population were principally selected based on their low HDL-C prior to statin medication as a predominant conventional lipid risk factor of their disease. In spite of this fact, many of the lipid ratios remained independently associated with statin-treated premature CHD in men adjusted for metabolic syndrome status or HDL-C in statistically limited analyses of this small population. As there was some resemblance to earlier reported plasma lipidomic findings, larger-scale systematic comparisons between plasma and HDL lipidome should be carried out in the future to elucidate their independent value as practically feasible and clinically beneficial classifying tools or prognostic markers in cardiometabolic disorders. Further research is required to illuminate the mechanistic implications of the lipidomic associations described in this study. 

## Figures and Tables

**Figure 1 ijms-22-04908-f001:**
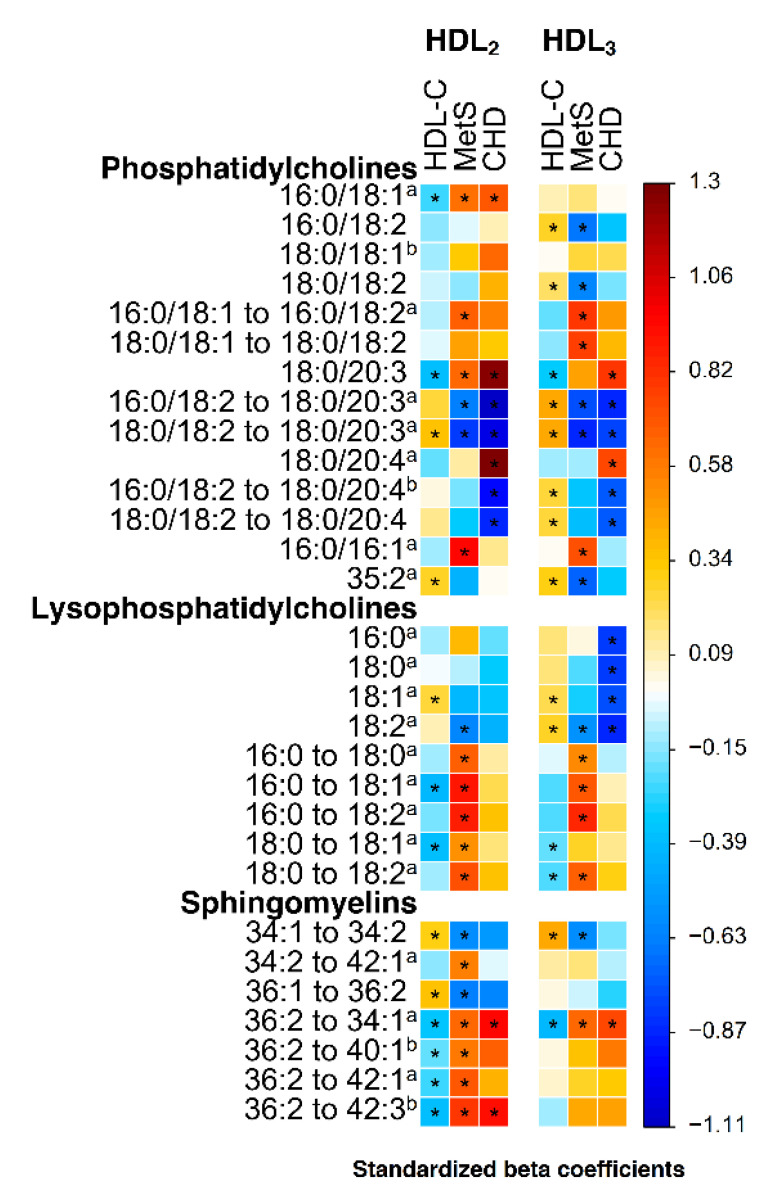
Associations of lipids and lipid ratios with metabolic syndrome (MetS), plasma HDL cholesterol level (HDL-C) or with premature coronary heart disease (CHD). Standardized beta coefficients of statistical models which estimated separately the effect of metabolic syndrome, premature CHD or HDL-C level on each of the lipid parameters are displayed. The models of metabolic syndrome and HDL-C were statistically adjusted for sex and age (premature CHD was investigated in manually age-adjusted groups of only male subjects, see Section 4.1 for details). Generalized estimating equation was applied in the modeling (see Section 4.2.7. for details). For numeric and further data, please see Appendix A. * Statistically significant association, Benjamini–Hochberg corrected false detection rate in multiple comparisons < 5% (see Section 4.2.7. for details); ^a^ log-transformed, ^b^ square root-transformed.

**Table 1 ijms-22-04908-t001:** Clinical characteristics and biochemical parameters of study subjects.

Characteristics	All Subjects			Men		Without Statin Medication or CHD	
	All	MetS	No MetS	Premature CHD ^1^	No CHD ^1^	MetS	No MetS
N	112	58	54	30	20	26	40
Age (y)	51 (40–58)	55 (47–60) ^4^	45 (32–53)	52 (49–57)	52 (44–59)	50 (40–60) ^4^	42 (30–51)
Women (*n*)	54 (48)	25 (43)	29 (54)	0 (0)	0 (0)	16 (62)	25 (63)
CHD (*n*)	39 (35)	29 (50) ^4^	10 (19)	30 (100)	0 (0)	0 (0)	0 (0)
Premature CHD (*n*)	35 (31)	26 (45) ^4^	9 (17)	30 (100)	0 (0)	0	0
Age at CHD onset (y)	50 (43–55)	51 (44–55)	47 (41–56)	47 (42–51)	NA	NA	NA
MetS (*n*)	58 (52)	58 (100)	0 (0)	22 (73)	10 (50)	26	40
Diabetes (*n*)	10 (9)	7 (12)	3 (6)	2 (7)	1 (5)	1 (4)	1 (3)
Hypertension (*n*)	38 (34)	31 (53) ^4^	7 (13)	14 (47) ^4^	1 (5)	10 (39) ^4^	2 (5)
BMI (kg/m^2^)	27.9 (24.9–30.5)	29.6 (27.8–31.6) ^4^	25.1 (22.2–27.8)	29.5 (27.5–31.2) ^4^	27.3 (25.1–28.4)	28.6 (26.2–30.6) ^4^	25.1 (21.3–27.9)
Waist (cm)	94 (85–102)	101 (95–106) ^4^	88 (80–92)	102 (97–107) ^4^	95 (88–99)	95 (88–100) ^4^	86 (77–92)
Syst. BP (mm Hg)	128 (115–141)	132 (121–142) ^4^	122 (113–135)	130 (115–140)	130 (120–141)	131 (119–142)	121 (113–134)
ACE/ATII (*n*)	36 (32)	27 (47) ^4^	9 (17)	17 (57) ^4^	1 (5)	6 (23)	2 (5)
Statin (*n*)	42 (38)	29 (50) ^4^	13 (24)	27 (90) ^4^	1 (5)	0 (0)	0 (0)
Smoker (*n*)	29 (26)	17 (29)	12 (22)	10 (33)	5 (25)	7 (27)	10 (25)
Ex-smoker (*n*)	32 (29)	18 (31)	14 (26)	16 (53)	7 (35)	6 (23)	7 (18)
Pack-years ^2^	1 (0–15)	10 (0–26) ^4^	0 (0–10)	23 (9–30) ^4^	10 (0–15)	0 (0–12)	0 (0–6)
Alcohol (doses ^3^/week)	2 (1–6)	2 (0–4)	2 (1–7)	3 (0–6) ^4^	5 (2–13)	3 (1–9)	2 (1–7)
Total C (mmol/L)	4.50 (3.93–5.20)	4.50 (3.90–5.38)	4.60 (4.00–5.05)	3.95 (3.48–4.40) ^4^	5.20 (4.93–5.98)	5.40 (4.43–6.00)	4.90 (4.30–5.30)
HDL-C (mmol/L)	1.28 (1.08–1.54)	1.14 (0.98–1.37) ^4^	1.41 (1.23–1.77)	1.07 (0.85–1.17) ^4^	1.28 (1.08–1.67)	1.26 (1.05–1.55) ^4^	1.51 (1.26–1.81)
LDL-C (mmol/L)	2.70 (2.30–3.30)	2.70 (2.30–3.48)	2.75 (2.28–3.23)	2.40 (1.95–2.90) ^4^	3.30 (2.90–4.00)	3.35 (2.68–4.13)	2.90 (2.53–3.50)
TG (mmol/L)	1.12 (0.84–1.73)	1.48 (1.10–2.13) ^4^	0.90 (0.64–1.11)	1.37 (0.97–2.24)	1.38 (0.63–2.14)	1.71 (1.06–2.13) ^4^	0.89 (0.63–1.08)
FFA (mmol/L)	0.51 (0.37–0.65)	0.49 (0.41–0.63)	0.53 (0.36–0.69)	0.41 (0.32–0.54)	0.46 (0.35–0.57)	0.55 (0.44–0.69)	0.53 (0.37–0.68)
Glucose (mmol/L)	5.8 (5.4–6.5)	6.2 (5.7–6.6) ^4^	5.4 (5.1–5.9)	6.3 (5.7–6.7)	5.8 (5.5–6.4)	6.0 (5.7–6.4) ^4^	5.4 (5.1–5.7)
HOMA-IR	2.1 (1.3–3.6)	2.7 (2.0–4.5) ^4^	1.4 (1.1–2.3)	3.2 (2.0–4.5) ^4^	1.5 (1.1–2.0)	2.0 (1.5–2.8) ^4^	1.3 (1.0–2.1)
Total adip. (mg/L)	7.6 (5.3–10.6)	6.2 (4.5–8.9) ^4^	8.7 (6.9–12.8)	5.3 (3.9–7.3) ^4^	6.8 (4.9–10.2)	7.4 (5.2–12.2)	8.2 (6.8–12.6)
HMW adip. (mg/L)	2.4 (1.3–4.6)	1.7 (1.1–3.2) ^4^	3.1 (1.9–5.7)	1.4 (1.1–2.6)	1.9 (1.1–4.0)	2.2 (1.2–4.4)	3.0 (1.9–5.5)
ALT (U/L)	25 (17–33)	29 (21–38) ^4^	22 (16–29)	31 (25–41)	27 (20–34)	28 (19–34) ^4^	19 (14–27)
Creatinine (µmol/L)	66 (59–72)	66 (58–72)	66 (61–71)	69 (61–74)	72 (66–74)	63 (56–71)	64 (59–70)

Values are expressed as a median (interquartile range) or as a number of subjects (percentage). Abbreviations: CHD, coronary heart disease; MetS, metabolic syndrome; ACE/ATII, angiotensin-converting enzyme inhibitor or angiotensin receptor II blocker medication; waist, waist circumference; syst. BP, systolic blood pressure; NA, not applicable; Total C, total cholesterol; HDL-C, HDL cholesterol; LDL-C, LDL cholesterol; TG, triglycerides; FFA, free fatty acids; adip., adiponectin; HMW, high-molecular-weight. ^1^ Only subjects over 35 years of age included in age-matched groups; ^2^ pack-year = 20 cigarettes/day during one year; ^3^ dose = 12 g of alcohol; Mann–Whitney U-test, Pearson chi-squared test or Fisher’s test between premature CHD/no CHD or between MetS/no MetS: ^4^ *p* < 0.05.

**Table 2 ijms-22-04908-t002:** Associations between selected phosphatidylcholine or sphingomyelin ratios and premature coronary heart disease (CHD) status adjusted for metabolic syndrome (MetS) status, parameters of MetS or cardiovascular risk factors.

Adjusted for		Phosphatidylcholines						Sphingomyelins			
		16:0/18:2 to 18:0/20:3		18:0/18:2 to 18:0/20:3		16:0/18:2 to 18:0/20:4		36:2 to 34:1		36:2 to 42:3	
		HDL_2_ ^4^	HDL_3_ ^4^	HDL_2_ ^4^	HDL_3_ ^4^	HDL_2_ ^5^	HDL_3_ ^5^	HDL_2_ ^4^	HDL_3_ ^4^	HDL_2_ ^5^	HDL_3_ ^4^
MetS	β_1_	−1.06 ^3^	−0.79 ^2^	−0.91 ^3^	−0.67 ^1^	−1.01 ^3^	−0.80 ^2^	0.91 ^2^	0.66 ^1^	0.74 ^3^	0.37
	CI	−1.53–−0.59	−1.30–−0.28	−1.37–−0.46	−1.19–−0.15	−1.52–−0.51	−1.32–−0.27	0.40–1.43	0.14–1.17	0.34–1.15	−0.20 –0.94
	β_2_	−0.21	−0.30	−0.46	−0.50	0.28	0.18	0.13	0.32	0.72 ^3^	0.40
	CI	−0.73–0.30	−0.83 –0.23	−0.96–0.03	−1.06–0.06	−0.31–0.88	−0.36–0.73	−0.33–0.58	−0.21–0.84	0.33–1.11	−0.15–0.95
HDL-C ^4^	β_1_	−0.83 ^2^	−0.49	−0.63 ^2^	−0.35	−1.17 ^3^	−0.70 ^1^	0.84 ^2^	0.36	0.68 ^1^	0.38
	CI	−1.31–−0.35	−1.03–−0.04	−1.06–−0.20	−0.90–0.20	−1.77–−0.58	−1.25–−0.15	0.24–1.43	−0.20–0.91	0.11–1.25	−0.24–1.01
TG ^5^	β_1_	−1.05 ^3^	−0.79 ^2^	−0.93 ^3^	−0.68 ^2^	−1.01 ^3^	−0.77 ^2^	0.90 ^2^	0.66 ^1^	0.80 ^3^	0.42
	CI	−1.53–−0.58	−1.29–−0.29	−1.39–−0.46	−1.20–−0.17	−1.47–−0.54	−1.27–−0.27	0.37–1.42	0.15–1.18	0.36–1.23	−0.18–1.01
VLDL-TG ^5^	β_1_	−1.02 ^3^	−0.75 ^2^	−0.88 ^3^	−0.63 ^1^	−1.03 ^3^	−0.77 ^2^	0.88 ^2^	0.63 ^1^	0.77 ^2^	0.40
	CI	−1.48–−0.55	−1.26–−0.24	−1.34–−0.42	−1.15–−0.11	−1.50–−0.56	−1.27–−0.27	0.34–1.41	0.11–1.15	0.30–1.24	−0.21–1.01
VLDL protein ^5^	β_1_	−1.00 ^3^	−0.74 ^2^	−0.86 ^3^	−0.63 ^1^	−1.03 ^3^	−0.75 ^2^	0.86 ^2^	0.63 ^1^	0.77 ^2^	0.39
	CI	−1.46–−0.55	−1.22–−0.25	−1.30–−0.43	−1.11–−0.14	−1.51–−0.55	−1.25–−0.25	0.34–1.38	0.12–1.13	0.32–1.22	−0.20–0.97
Adipon. ^4^	β_1_	−0.96 ^3^	−0.71 ^2^	−0.86 ^2^	−0.60 ^1^	−1.00 ^3^	−0.74 ^2^	0.90 ^2^	0.61 ^1^	0.82 ^2^	0.44
	CI	−1.42–−0.50	−1.23–−0.19	−1.35–−0.37	−1.13–−0.06	−1.47–−0.53	−1.24–−0.25	0.32–1.48	0.08–1.14	0.24–1.40	−0.16–1.03
HMW adipon. ^4^	β_1_	−0.97 ^3^	−0.66 ^1^	−0.87 ^3^	−0.62 ^1^	−0.98 ^3^	−0.70 ^2^	0.86 ^2^	0.75 ^2^	0.82 ^2^	0.48
	CI	−1.44–−0.51	−1.17–−0.15	−1.35–−0.39	−1.14–−0.10	−1.47–−0.48	−1.22–−0.18	0.31–1.42	0.24–1.25	0.28–1.35	−0.11–1.06
Waist ^5^	β_1_	−0.97 ^3^	−0.73 ^2^	−0.83 ^3^	−0.58 ^1^	−0.94 ^3^	−0.77 ^2^	0.88 ^3^	0.66 ^1^	0.59 ^2^	0.37
	CI	−1.41–−0.52	−1.22–−0.24	−1.29–−0.37	−1.09–−0.07	−1.46–−0.42	−1.34–−0.20	0.39–1.37	0.13–1.18	0.17–1.01	−0.24–0.98
HOMA-IR ^4^	β_1_	−0.99 ^2^	−0.71 ^1^	−0.85 ^2^	−0.63	−1.10 ^3^	−0.87 ^2^	0.92 ^2^	0.84 ^2^	0.55 ^1^	0.44
	CI	−1.55–−0.42	−1.29–−0.12	−1.40–−0.30	−1.25–0.00	−1.64–−0.57	−1.41–−0.33	0.39–1.46	0.24–1.45	0.04–1.07	−0.24–1.12
Smoker	β_1_	−1.06 ^3^	−0.82 ^2^	−0.99 ^3^	−0.75 ^2^	−0.92 ^3^	−0.72 ^2^	0.96 ^3^	0.75 ^2^	0.91 ^3^	0.48
	CI	−1.53–−0.59	−1.30–−0.33	−1.49–−0.49	−1.26–−0.23	−1.37–−0.48	−1.20–−0.24	0.45–1.47	0.22–1.29	0.42–1.39	−0.09–1.05

The following statistical models (generalized estimating equation, see Section 4.2.7. for details) estimate the effect of premature CHD on lipid parameters after adjusting separately for each of the confounding factors among men adjusted manually for age (for details of age adjustment see Section 2.1): *z*-score (sex-specific) of the lipid variable = β_1_ x premature CHD (coded as 0 without CHD or 1 with premature CHD) + β_2_ x the adjusting variable (sex-specific *z*-score if continuous) + the intercept. A unit of *z*-score is one standard deviation of native or normalized (transformed) lipid variable. CI, 95% confidence interval. Adjusting variables: MetS, metabolic syndrome status coded as 0 without metabolic syndrome or 1 with metabolic syndrome; HDL-C, plasma HDL cholesterol; TG, total plasma triglycerides; VLDL-TG, plasma VLDL triglycerides; VLDL protein, plasma VLDL protein; adipon., total plasma adiponectin; HMW-adipon., plasma high-molecular-weight adiponectin; waist, waist circumference; HOMA-IR, HOMA index of insulin resistance; smoker = 1 or non-smoker = 0 in the models. Unadjusted *p*-value of β: ^1^ *p* < 0.05, ^2^ *p* < 0.01, ^3^ *p* < 0.001; ^4^ log-transformed, ^5^ square root-transformed.

## Data Availability

The data presented in this study are available in the Appendix A to this manuscript and at https://doi.org/10.1371/journal.pone.0171993.s003.

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
