# Peer review of "Distinct Fatty Acid Compositions of HDL Phospholipids Are Characteristic of Metabolic Syndrome and Premature Coronary Heart Disease—Family Study"

_ijms, 2021, doi:10.3390/ijms22094908_

Round 1

Reviewer 1 Report

Title: Distinct fatty acid compositions of HDL phospholipids are characteristic of metabolic syndrome and premature coronary heart disease

Paavola T., Bergmann U., Kuusisto S., Kakko S., Savolainen M.J., Salonurmi T.

Manuscript ID: ijms-1187682

Objective:

In the current manuscript, the authors investigated the lipidome of HDL2 and HDL3 subfractions with special attention to phosphatidylcholine (PC), lysophosphatidylcholine (LPC), and sphingomyelin (SM) to evaluate associations with premature coronary heart disease and metabolic syndrome (MetS) in families, where low HDL-C predisposes premature CHD. They observed, that LPC was modified in MetS with depletion of LPC linoleic acid relative to palmitic acid. Furthermore, PCs containing oleic acid increased at the expense of PCs with linoleic acid, i.e., a surplus in the ratios of PC (16:0/18:1) to (16:0/18:2). They concluded, that these changes in the fatty acid pattern of LPCs and PCs are characteristic for MetS and that certain PC and SM ratios are related consistently to premature CHD and MetS.

Points of criticism:

First of all, I would like to congratulate the authors on this detailed study. There is almost too much information, which sometimes leads to a loss of coherence so that the manuscript seems disjointed.

The main finding, that linoleic acid depletion in LPC/PC was, unfortunately not accompanied by the measurement of oxidative stress and antioxidants. A retrospective measurement cannot be made in this regard, especially since the samples were taken between 2007 and 2009 - however, one should point out this connection with recent literature (recommendation: Stadler et al. 2021 - see “literature”).

Besides, the question arises if this effect is of the dietary origin or connected to oxidative stress – especially considering that CHD male subjects displayed heavier exposure to smoking. Also, therapeutic approaches (e.g., supplementation with polyunsaturated fatty acids and antioxidants) should be discussed – although a definitive conclusion was not possible based on the presented data.

The authors have very purposefully emphasized the limitations of the paper, e.g., the confounding effect of statin medication. Thus, treated subjects may no longer be compared with untreated subjects – including inconsistencies between MetS and CHD, which complicates the content even more. The core statements for each group should be presented in an accentuated manner, e.g., with special attention to the effects (positive and negative) of statin treatment about to the lipidome of HDL, e.g., concerning HDL-C concentrations in MetS (without statin medication or CHD), which were significantly increased.  

Referring to the fact that only men were used for CHD because women generally have a better constitution, could be corroborated by a reference (recommendation: Wonisch et al. 2012) - see "Literature". Besides, the “family character” of this study was presumably the reason for the missing age-matched controls in the case of CHD.

Minor points:

Title:

The study setting as a “family study” should be emphasized already in the title.  

Material and Methods:

4.2.2.

Blood sampling should be described in detail (e.g., forearm, sitting position, …).

4.2.3.

The statement for manufacturer and Country needs to be completed for each parameter.

There are several typing errors (especially blank and hyphen) throughout the manuscript that need to be improved.

The formatting of Table 1 should be improved.

References:

Ref.5 – Kontush et al. 2015 was cited in PubMed – Handb Exp Pharmacol 224: 3-51

Recommended Literature:

Two papers that might augment the current study:

Obesity affects HDL Metabolism, composition and subclass distribution. Stadler J.T. et al.; Biomedicines 2021,9,242. https://doi.org/10.3390/biomedicines9030242

Oxidative stress increases continuously with BMI and age with unfavourable profiles in males. Wonisch et al. The Aging Male 2012; 15(3):159-65

Reviewer 2 Report

The manuscript of Paavola et al describes studies on how the lipidome of the main phosphatidylcholine (PC), lysophosphatidylcholine (LPC) and sphingomyelin (SM) species of HDL2 and HDL3 subfractions is associated with premature coronary heart disease (CHD) or metabolic syndrome (MetS) in families where common low HDL-C predisposes to premature CHD. Liquid chromatography-mass spectrometry  (LC-MS) was used to analyze the lipidome. Alterations in particular fatty acid composition of certain LPCs and PCs displayed a characteristic pattern in MetS, enriched with palmitic, palmitoleic or oleic acids relative to linoleic acid. In addition, certain PC and SM ratios related consistently to CHD and MetS.

The results are interesting and I suggest publication, however after a revision.

Comments

  1. Either in introduction or in discussion the authors have to discuss some very recent papers. “Exploratory analysis of large-scale lipidome in large cohorts: are we any closer of finding lipid-based markers suitable for CVD risk stratification and management?” Analytica Chimica Acta Volume 1142, 15 January 2021, Pages 189-200. “Alterations of endogenous sphingolipid metabolism in cardiometabolic diseases: Towards novel therapeutic approaches.” Biochimie 2020 169, pp. 133-143. “Structure-function relationships of HDL in diabetes and coronary heart disease”. JCI Insight 2020 5(1),e131491. “Sphingolipids as biomarkers of disease”. Advances in Experimental Medicine and Biology 2019 1159, pp. 109-138.
  2. As recently shown (The HDL lipidome is widely remodeled by fast food versus Mediterranean diet in 4 days. Metabolomics 2019, 15(8),114), the diet is an important parameter that easily affects the lipidome. This parameter has to be discussed in the Discussion.
  3. Could the authors make a correlation between the free fatty acid levels and composition of each particular fatty acid, which is altered in LPCs, PCs, SMs?
